# From Complex Interventions to Complex Systems: Using Social Network Analysis to Understand School Engagement with Health and Wellbeing

**DOI:** 10.3390/ijerph16101694

**Published:** 2019-05-14

**Authors:** Hannah J. Littlecott, Graham F. Moore, Hugh Colin Gallagher, Simon Murphy

**Affiliations:** 1DECIPHer, UKCRC Centre of Excellence, Cardiff University, 1-3 Museum Place, Cardiff CF10 3BD, UK; MooreG@cardiff.ac.uk (G.F.M.); murphys7@cardiff.ac.uk (S.M.); 2Centre for Transformative Innovation, Faculty of Business and Law, Swinburne University of Technology, AGSE121 Hawthorne Campus, Melbourne, PO Box 218, Australia; cgallagher@swin.edu.au

**Keywords:** school health, complexity, complex systems, network, ego network analysis, social network analysis

## Abstract

Challenges in changing school system functioning to orient them towards health are commonly underestimated. Understanding the social interactions of school staff from a complex systems perspective may provide valuable insight into how system dynamics may impede or facilitate the promotion of health and wellbeing. Ego social network analysis was employed with wellbeing leads within four diverse case study schools to identify variability in embeddedness of health and wellbeing roles. This variation, as well as the broader context, was then explored through semi-structured qualitative interviews with school staff and a Healthy Schools Coordinator, sampled from the wellbeing leads’ ego-networks. Networks varied in terms of perceived importance and frequency of interactions, centrality, brokerage and cliques. Case study schools that showed higher engagement with health and wellbeing had highly organised, distributed leadership structures, dedicated wellbeing roles, senior leadership support and outside agencies embedded within school systems. Allocation of responsibility for wellbeing to a member of the senior leadership team alongside a distributed leadership approach may facilitate the reorientation of school systems towards health and wellbeing. Ego-network analysis to understand variance in complex school system starting points could be replicated on a larger scale and utilised to design complex interventions.

## 1. Introduction

Youth is a period where many protective and risk behaviours are formed, and there is growing evidence that the school environment can affect health and wellbeing [1]. This focus on school environments, rather than individuals within the school, is in line the Ottawa Charter principles, which emphasise a need to support health within the settings of people’s everyday lives [2]. Moreover, healthy behaviours have been found to track into adulthood. Thus, intervening at this early age may increase the likelihood of positive health and wellbeing and decrease the risk of disease development, such as cancer or coronary heart disease [3,4]. 

School health improvement can be defined as including all actions, such as policies and practices, employed with the aim of improving the health and wellbeing of students [5]. While health education dominated much early school health work, recent systematic reviews highlight the greater effectiveness of multi-level “complex interventions”, such as those using the World Health Organisation’s Health Promoting Schools (HPS) approach (i.e., combining curriculum development, environmental change and family engagement) to develop and implement health improvement activity [6]. While adding health topics to the curriculum is relatively straightforward, more complex multi-level interventions have often proven challenging to implement [7], perhaps tending to achieve minimal disruption to school system functioning and entrenched patterns of health and inequality, as a consequence [8]. There remains a substantial need therefore to better understand the challenges in achieving change within schools to overcome the negligible, modest or a lack of effects which has historically been found within many large-scale trials of interventions targeting school health [9].

Recent years have seen increasing movement toward viewing complex interventions not simply as the ‘installation’ of something new into a system, but as events within systems [10]. From this perspective complexity of the system, and of change efforts within it, are foregrounded as primary foci of study, while interventions cannot be described in isolation from the contexts they attempt to alter. Consistent with this perspective, one rapidly growing field of inquiry within public health which attempts to theorise challenges in system change, is complex adaptive systems (CAS) thinking [11]. A CAS is a dynamic network of many diverse agents and characteristics, acting and reacting to other agents’ behaviour, generating emergent system characteristics, which in turn exert influence on individual behaviour [11].

According to Hawe [10], complex systems are comprised of activity settings, the social networks between them, and time. Social dynamics within and between a systems’ activity settings may have the potential to improve or harm health. Thus, an intervention within a system can attempt to alter pre-existing social dynamics in order to achieve change and enhance the health promoting potential of the system, through displacing problematic practices and introducing new ones. While health education often focuses on one activity setting (altering classroom dynamics), whole school interventions based upon the HPS approach target various activity settings within schools, altering social dynamics within and between activity settings to promote health. Interventions may involve efforts to fill structural holes, where a lack of brokerage exists between different cliques with distinct resources and information, through the creation of new activity settings to promote interaction between stakeholders, such as school staff and parents [12]. A focal point of this is school staff, due to a need to understand and identify the system characteristics which may facilitate or hinder the implementation of any new intervention, or system disruption, that is introduced as new policies, practices and ways of working flow into school systems via staff networks [13].

CAS perspectives have gained much influence in school health research [14,15], and public health science more broadly in recent years [16,17]. Indeed, attempts have been made to operationalise the interaction between context and school-based interventions through conceptualising these as network systems comprised of human and non-human entities [18]. A qualitative study of whole school approaches in Australia, where interviews were conducted with secondary school staff, found that schools possess many characteristics of CAS; they comprise diverse and ever-changing agents, are nested within supra-systems, such as Local Education Authorities, and comprise numerous subsystems [11]. Outcomes produced by schools are influenced by diverse interactions among agents, such as staff, within and between schools, as well as with communities and families [7]. Thus, the process of orienting schools towards supporting health and wellbeing centres around diverse interactions among staff and between staff and other stakeholders, such as students, parents and outside agencies, regarding health and wellbeing.

Schools have well-developed mechanisms for prioritising information related to their ‘core business’ of educational attainment, while internal and external monitoring structures provide feedback loops and inform subsequent practice in relation to education [11]. Due to poor implementation, the Health Promoting Schools approach provides evidence of a somewhat naïve approach to the design, implementation and evaluation of interventions within complex systems, whereby the impact on ‘core business’ of educating students is often overlooked or ignored. This further supports a focus on system organisation and the staff who will be implementing health improvement activity in schools. For example, introducing a disruption to the system in the form of a health intervention without engaging with or understanding current system functioning may lead to the triggering of self-organisation processes which work to return the system to order and ‘wash out’ the new intervention. This highlights an implementation gap, whereby health interventions fail to be implemented into the reality of the school setting [19].

A commonly used means for conceptualising and measuring the dynamics of social systems is through social network analysis (SNA). Social networks are webs of social ties, which link people together. Most commonly, these are one-to-one links, conceptualised in terms of interaction (i.e., direct communication), affective ties (e.g., liking/disliking), role relationships (e.g., kinship), or various ideas of social exchange (e.g., social support) [20]. Within schools, interpersonal interactions and relationships are shaped not only by individual characteristics of agents (e.g., preferences, choices, and motivations), but also by characteristics and institutional practices of the school. Moreover, schools comprise activity settings and interactions within and between them. Social networks are influenced by formal hierarchal structure, but also often comprise subgroups that deviate from this, and may differ across schools and activity settings [21]. Understanding the functioning of these networks, through social network methods, underpinned by a complex systems perspective, is a potentially valuable and underused approach for understanding how existing social structures and practices (e.g., the extent to which, and ways in which, school staff interact with each other and other agents within and outside of the school system) impede or facilitate the orientation of school systems toward health. 

To date, whole network analysis has been used with secondary school staff, to identify key players within the school for implementing complex interventions [22]. It may also be useful in understanding how health improvement information flows through school systems [11], and in planning system-level interventions [23]. While useful in understanding internal school structure, there is an inherent tension between the need to bound the system at the school gates to conduct whole network analysis, and the conceptualisation of schools as complex systems with highly permeable boundaries [24]. Health improvement will often require close partnership with outside agencies [25], and often aims explicitly to enhance relationships with groups beyond the school gates, such as parents and community groups. Although sacrificing some understanding of the structural characteristics of school systems, ego-network analysis with key individuals can provide an understanding of health-related networks within the school of individuals with responsibility for health improvement, and to simultaneously capture interactions with external systems [21]. This method has been used in multiple case studies focusing on educational teacher networks [26]. No studies have to date used ego network analysis to investigate interactions related to health improvement, although its usefulness in capturing interactions across boundaries was demonstrated by a study of primary school communications between teachers and outside agencies [27].

In Wales, within most schools, the role of coordinating health and wellbeing improvement activities is allocated to a member of teaching staff (referred to throughout as wellbeing leads). Previous studies have demonstrated that student health outcomes can be influenced by having an individual dedicated to health improvement [28,29]. Investigating the ego networks of these wellbeing leads, combined with qualitative interviews, is likely to yield rich data regarding the embeddedness of this role into the Welsh school system [30]. These individuals are likely to undertake a brokerage role between external agencies and school stakeholders, therefore acting as a key individual in the diffusion of a health and wellbeing intervention throughout the system. The success of this diffusion may depend on the characteristics of school staff networks and these individuals’ position within them, as well as their level of connectedness with outside agencies. This paper first uses social network analyses within four diverse case study schools to measure wellbeing leads’ ego networks and determine the extent to which health and wellbeing roles are embedded into school systems. Next, semi-structured qualitative interviews are used to explore the broader school context surrounding these network structures. This paper will aim to increase understanding of how variability in network structures, and the positions of key change agents within these, may facilitate or impede attempts to orient school systems toward health and wellbeing.

## 2. Materials and Methods

A brief summary of the data sources used in this study is displayed in Table 1.

### 2.1. Study Design

The research design consisted of four in-depth, mixed method school case studies involving ego social network analysis and semi-structured interviews with school staff. The research was conducted in Wales, United Kingdom between September 2014 and June 2015.

### 2.2. Ethical Approval

Ethical approval was obtained by Cardiff University’s School of Social Science Research Ethics Committee in May 2014 (2015PHW0011) and a Disclosure and Barring Service (DBS) check was undertaken by the researcher in December 2013. Additional Research and Development approval was obtained from the National Health Service (SREC-1247). This was required to conduct an interview with a Healthy School Coordinator who was employed by their Local Health Board, encompassed by the National Health Service. Written informed consent was obtained for all interviews.

### 2.3. Case Study Sampling

Purposive sampling using replication logic was employed to select four case study schools, sampled to represent different geographical locations, sizes, socioeconomic status (SES) and stages of the Health Promoting Schools Scheme. Schools were approached via a telephone call, repeated on a weekly basis until the relevant person was reached or a definitive answer was provided regarding participation. Out of the eight schools contacted by telephone, two agreed to participate. A group email invitation was then sent out to all schools who were members of the School Health Research Network (referred to as the ‘research network’ throughout). Three schools replied within two hours to express an interest in participating. The sampling criteria were applied again to recruit third and fourth case study. Pseudonyms were used throughout this manuscript to protect the anonymity of participating case study schools. Case study characteristics are summarised in Table 2.

The research network is a Welsh infrastructure for school-based health improvement research. It conducts biannual surveys with member secondary schools in Wales and provides bespoke pupil health and wellbeing reports to schools outlining statistics relating to their pupils’ levels of diet, physical activity, substance use, mental health and wellbeing, as well as a broad range of other school engagement action [15,31]. In 2014, there were 69 member schools [5].

### 2.4. Case Study Schools’ Engagement with the School Health Research Network

This research was undertaken within the context of the School Health Research Network. Data collected as part of the research network surveys were used to contextualise case study data within the current study. Background information on each school was derived from routine data sources and from a survey within the research network, collected in between September 2014−June 2015, regarding school engagement with feedback provided in the form of individualised wellbeing reports of pupil health behaviours by the research network. At the time of the survey the research network had a membership of 69 schools, 34 of which completed the survey. The survey measured the extent to which schools had attended research network events, received and read their feedback reports, discussed and intended to discuss the results with stakeholders and intended to take action from the results. As a result, the case study schools were ranked highest to lowest according to their level of engagement with the research network and represented a continuum of this engagement; Highbridge School was ranked 1, Oakwood School 2, Woodlands School 3 and Greenfield School 4.

### 2.5. The Embeddedness of Health Improvement within Case Study Schools

To further contextualise case study data, background information on each school was derived from a School Environment Questionnaire, collected by the research network in early 2016. These school environment data were collected to analyse the context of schools within the research network between March and May 2016. Out of 115 member schools, a response was received from 100 schools, a response rate of 87%. 

An indicator of the embeddedness of health improvement within case study schools, created by the research network team, was used. The indicator of embeddedness of health improvement related to the three topics within the HPS Scheme [2]: Curriculum, environment (measured by a number of policies related to health) and parental involvement. Firstly, schools indicated to which year groups, and in which subjects, the following topics were taught; healthy eating, physical activity, tobacco education, drug education, alcohol education, mental health and wellbeing and sex and relationships education. Sum scores for each individual topic were created and subjected to factor analysis; all health topics, apart from mental health, loaded onto one factor. Therefore, mental health in the curriculum was taken forward as its own variable whilst the other health topics were combined to generate physical health in the curriculum variable. These scores ranged from 0–10 for mental health and 11–75 for physical health in the curriculum.

Next schools were asked whether they had a written policy for smoking, drugs, alcohol, healthy eating, mental health, violence against women and suicide prevention, with a score generated for each school indicating the number of health topic areas covered by a written policy (ranging from 0–7). Finally, three parental involvement in decision-making questions, the estimated proportion of parents involved in health improvement, the number of areas in which parents were involved and the number of mechanisms (such as PTA groups) through which parents were involved, were combined into a single variable as factor analysis found these three questions to load onto a single factor. Individual subcomponents were scaled to represent scores from 0–1 and then combined to create a composite score of overall embeddedness of health in the school. This resulted in possible scores of 0 (lowest possible embeddedness) to 3 (highest possible). As a result, the case study schools were ranked highest to lowest according to their level of embeddedness of health; Highbridge School was ranked 1, Woodlands School 2, Greenfield School 3 and Oakwood School 4.

### 2.6. Social Network Analysis with Wellbeing Leads

Ego network analysis, whereby the perceptions of a focal participant of their immediate social network and their embeddedness in their social environment is measured [34], was employed with the wellbeing lead within each case study during a qualitative interview. Ego network data were collected via a physical visualisation method [30]. Interviewees (“egos”) underwent name-generation via free recall [30]. Participants were asked to list names and job titles of all individuals (“alters”) with whom they routinely interacted regarding health improvement, within and outside of the immediate school setting.

The ego was asked to use different coloured post-it notes according to stakeholder group (i.e., Senior Leadership Team (SLT), teaching staff, non-teaching staff, parents/students and individuals/organisations external to the school). Egos then assigned the following attributes to alters by marking numbers onto each post-it note according to a key; age group, gender, frequency of interaction and length of service. After this, egos were asked to draw lines between alters to represent whether these alters interacted with each other in relation to health and wellbeing and to indicate the importance of each interaction for school health and wellbeing. Examples of such interactions include planning a new health improvement activity, such as a new physical activity programme or sharing information about pupils experiencing challenges, such as poor mental health.

Egonet software was used to conduct all statistical analyses and to create diagrammatic representations (net-maps) of each network. Betweenness centrality (brokerage) scores and a number of cliques were calculated for each ego network. Betweenness centrality is a measure of brokerage, measuring whether alters sit on the shortest path between other nodes (the individuals or groups that are nominated as part of a network) [34,35]. Brokers may mediate between informal subgroupings, which can be observed in the form of cliques [36]. A clique is a subset of three or more alters who are all connected to one another, where no other alter is connected to all members [37]. Cliques may indicate the presence of a small shared group setting in which more than two people interact.

### 2.7. Qualitative Interviews with Wellbeing Leads and Other School Staff

The ego-network analysis was embedded within face to face, semi-structured interviews with each wellbeing lead. Results of the wellbeing leads’ ego social network analysis were then utilised to sample key informants, at varying levels of proximity to the ego and involvement in health within the school, to participate in further interviews. Participants included four to five members of staff or Healthy School Coordinators per school, including the wellbeing lead, senior leadership, subject teachers, subject head teachers, support staff and Personal and Social Education (PSE) staff. Staff were purposively selected and recruited via a snowball sampling technique and approached by the wellbeing lead. Written informed consent was obtained prior to commencing each interview. See Table 3 for an overview of participant characteristics.

Interview questions were piloted with two individuals who work with schools, or work in schools that did not participate as a case study. Interview schedules were adapted throughout data collection to follow interesting leads from previous interviews. Interviews lasted between 30 min to one hour and were recorded using a Dictaphone and then transcribed. Notes were also taken throughout the interviews to record observations about the setting, participants’ attitudes and non-verbal communication.

Data from qualitative interviews focused on the advantages and disadvantages of allocating responsibility for health and wellbeing to a junior member of staff versus a member of the SLT. Perceptions of how to achieve a balance between these two options in order to enhance the team structure for health and wellbeing within a school was further explored.

### 2.8. Qualitative Interview Analysis

Coding was conducted using NVivo software (version 10, QSR International (UK) Limited, London, UK). Interviews were analysed using thematic analysis [38] with aspects of a Grounded Theory approach incorporated [39]. Inductive open coding was used to develop an initial coding system. The initial overarching themes were as follows; comparison with other schools, data usage, family’s role in health promotion, friends’ role in health promotion, health and wellbeing programmes in the school, interactions, job role, link between health and educational outcomes, perceptions of school health promotion, school ethos and social network analysis. Codes were then compared and structured further. This involved repeated reading of the transcripts in an active manner [38]. In line with grounded theory, a second scan of the interview transcripts was then undertaken, whilst actively suppressing any presuppositions about the data, to identify any other possible themes. All codes were then organised into overarching themes and sub-themes. Themes were then reviewed in terms of whether the data extracts fit into each coherent theme and whether the themes and sub-themes accurately represented the overall dataset. Alterations were made accordingly [38], before naming and defining the themes. This was an iterative process, whereby pertinent codes were elaborated upon within future interviews.

## 3. Results

The wellbeing leads for each case study school were female. For Greenfield school, the wellbeing lead was a physical education teacher aged 26–35 years who had been in post for 8 years. For Woodlands school, the wellbeing lead was an assistant head teacher and the wellbeing leads for Highbridge and Oakwood schools were deputy head teachers. Each of the wellbeing leads for Woodlands, Highbridge and Oakwood schools were aged 46–55 years and had been in post for >25 years.

### 3.1. Social Networks of Wellbeing Leads in Case Study Schools

#### 3.1.1. Influential Champions for Health: Characteristics and Position within Social Networks

Figure 1, Figure 2, Figure 3, Figure 4 and Figure 5 provide a key and net-map diagrams for the wellbeing leads’ ego network in each of the four case studies. The net-map diagrams visualise the individuals that the wellbeing lead or ‘ego’ has reported interacting with. Thus, the wellbeing lead is not included within the net-map as they are linked to every node. A Appendix A provides detailed keys for the job roles included within each net-map.

Highbridge School’s wellbeing lead reported a highly organised health and wellbeing-related team structure whereby members of the core health and wellbeing group included the safeguarding officer and head teacher, who were both members of senior leadership, and the wellbeing manager, who was a member of non-teaching staff, were present within most cliques. The core group had the highest betweenness centrality scores and acted as brokers between all other alters in the network, including outside agencies.

A limited health and wellbeing-related team structure was reported by the wellbeing lead in Oakwood School, consisting of four members of senior leadership (including the head teacher and deputy head teacher), heads of year, school nurse, additional learning needs coordinator and the head of PSE. 

Whilst Greenfield School’s wellbeing lead reported a limited team structure, the presence of a small group setting in one section of the health and wellbeing-related network is demonstrated by the fact that the assistant head for wellbeing is engaged in several cliques, mainly with non-teaching staff with dedicated wellbeing roles. Meanwhile the Assistant head for PSE engaged in several dyadic ties with teaching staff. This suggests one-to-one settings or interactions, perhaps eliciting information exchange with little collective consultation. No other members of senior leadership were included in the network.

Woodlands School’s wellbeing lead reported the least developed team structure in relation to health and wellbeing, whereby cliques were mainly comprised of homogenous groups, such as members of the SLT (including the head teacher, deputy head teachers and assistant head teachers), with limited connections and brokerage between them.

The only schools in which the head teacher played a key brokerage role in wellbeing leads’ health and wellbeing-related networks were Highbridge (engagement rank 1, embeddedness rank 1) and Oakwood (engagement rank 2, embeddedness rank 4) (see Table 4). In Oakwood School there were 22 cliques, compared to 19 in Woodlands School (engagement rank 3, embeddedness rank 2), 14 in Highbridge School and six in Greenfield School (engagement rank 4, embeddedness rank 3). 

These results demonstrate that case study schools’ network structures vary according to the allocation of responsibility for leading health and wellbeing, the extent to which SLT members play brokerage roles, the perceived importance and frequency of interactions with other key agents with regards to health and wellbeing, the number of roles relating to health, and the embeddedness of outside agencies into school systems. The qualitative results, presented below, will elaborate on the context surrounding these findings, as well as the perceived impact of the allocation of responsibility for leading health and wellbeing and the structure of wider leadership models.

#### 3.1.2. Frequency of Health and Wellbeing-Related Interactions and Their Importance Ratings

Table 5 and Table 6 detail the quantitative network characteristics for each of the four case studies. Overall, the wellbeing lead within Highbridge School (engagement rank 1, embeddedness rank 1) reported the highest proportion of interactions as extremely important (15/25; 60.0%), which included substantially more interactions with outside agencies (4/8; 80.0%) rated as ‘extremely important’ for school health than the other three case studies. The wellbeing lead for Woodlands School (engagement rank 3, embeddedness rank 2) did not rate any interactions with the SLT as important and Oakwood School (engagement rank 2, embeddedness rank 4) ranked 3/6 (50.0%) of members as extremely important. The wellbeing lead in Greenfield School (engagement rank 4, embeddedness rank 3) reported two out of two, the highest percentage (100.0%), but the smallest absolute number, of extremely important interactions with SLT members. This was closely followed by Highbridge School which reported four out of five (80.0%). Within Highbridge School, reported interactions about health improvement with outside agencies were more frequent compared to the other three case studies. In addition, interactions with parents, students and teaching staff were less frequent in Greenfield and Woodlands Schools, compared to Highbridge and Oakwood Schools. Woodlands School’s wellbeing lead reported the highest frequency of interaction with non-teaching staff, whilst the wellbeing leads from Greenfield, Oakwood and Woodlands Schools reported interacting with all SLT within their network about health improvement more than 2–3 times per week.

### 3.2. Qualitative Perceptions of the Broader Context Surrounding Health and Wellbeing in Case Study Schools

#### 3.2.1. Perceptions of the Allocation of Responsibility for Leading Health and Wellbeing

Allocating the role of wellbeing lead to a member of the SLT was perceived by most to be important for orienting the school system towards health improvement [8]. For example, this was perceived to facilitate the mobilisation of authority to respond quickly to changes in policies, make important decisions quickly, delegate tasks, deal with outside agencies and remove children from classes for appointments with outside agencies.


*“I think having [name] who is the deputy head and our Inclusion and Wellbeing Officer, **the fact that that’s from that level at the senior management level.** She drives this wellbeing ethos in our school (…)”*
*Oakwood School, school nurse*

The importance of placing this role within the SLT is supported by the social network brokerage findings, whereby the head teacher was included in the top five betweenness centrality scores (see Table 4) in those schools who had allocated this role to a deputy head. Whereas, where this role was allocated to a teacher, assistant head teachers had the highest betweenness scores and the wellbeing lead had no direct communication with the head teacher regarding health. This suggests a higher level of access to key decision-makers within the school when the role of wellbeing lead is allocated within the SLT. This may influence system functioning through improved efficiency in information flow between sub-systems, alongside having the seniority to implement new ideas [8].


*“The wellbeing lead has to be part of the Senior Management Team because the wellbeing lead could never ever just be a middle manager because **huge decisions have to be made and it’s got to be pushed right from the top down** (…)”*
*Oakwood School, wellbeing lead*

The Healthy Schools Coordinator working with Greenfield School (engagement rank 4, embeddedness rank 3) also perceived the SLT to have a greater influence over eliciting action from other agents within the school.


*“(…) generally if someone is saying ‘this is a good thing to do’ **then it’s more likely to have an impact if it’s someone in the Senior Management Team**, than it is if it’s someone who is just an ordinary teacher, generally.”*
*Greenfield, Healthy Schools Coordinator*

By contrast, teaching staff with seniority and power within their own departments, but who were not members of SLT, were perceived to have limited impact on the school system outside of these subsystems [8,11]. This is possibly due to structural holes that are not bridged by brokers between departmental cliques [12]. This is supported by the ego social network analysis, which showed that heads of department tended to interact with members of the SLT, but often not with agents within other departments.


*“(…) even for myself as a head of department it’s easy for me to make sure things are in place in my department, but **if I go out of my department to say ‘Oh can you do this, can you do that’, it’s very, it’s difficult** (…)”*
*Woodlands School, head of science and student voice*

The perception of the importance of the allocation of the role of wellbeing lead to the SLT was maintained by most, despite some staff acknowledging that individuals in the SLT have more demands on their time. Some suggested that, although the authority to make decisions and change the system is perceived as an important factor in allocation of the role of wellbeing lead, the ability to delegate tasks and leadership across several more junior members of staff was crucial to exerting agency and was seen as a more realistic way of understanding school improvement. 

However, there was not universal agreement, with some staff perceiving that, because of the high workload on the SLT, allocating the role of wellbeing lead to a member of staff with more time to dedicate to the role may be more beneficial. 


*“ (…) **it would be beneficial to have one person without all these other jobs to see to, to be you know solely dedicated**, that would obviously (…) be a positive, you know?”*
*Highbridge school, teaching assistant*

A similar view was expressed by an assistant head teacher in Greenfield School (engagement rank 4, embeddedness rank 3), where the role of wellbeing lead was allocated to a PE teacher, who was not convinced that the allocation of the role of wellbeing lead to senior leadership would equate to more authority within the school. He argued that the benefit would instead be obtained by the allocation to a more junior member of staff who could dedicate more resources to that specific role. 


*“(…) So I think it’s actually a real positive of where it’s sat at the moment (…) in line with heads of departments as well because and because **it becomes that person’s primary driver and therefore it probably has more effects than it being part of wider job brief higher up I think.**”*
*Greenfield School, assistant head for PSE*

#### 3.2.2. Wider Leadership Models

Here, a comprehensive team structure refers to the extent to which health-related roles are embedded in the school system. Within the case studies, this was generally perceived by school staff to be a further important characteristic for creating a school system that is conducive to health improvement. School wellbeing teams were reported to comprise several non-teaching staff with dedicated wellbeing roles.

The wellbeing lead from Woodlands School (engagement rank 3, embeddedness rank 2) reported that the difficulty prioritising health and wellbeing, reported in the previous section, may be due to the minimal team structure for wellbeing within that school. Whilst other members of staff expressed the need for one individual to have overall responsibility, while delegating to a team.


*“Absolutely, I couldn’t do my job if it wasn’t for the fact that I had a member of senior team who, sometimes she will say, ‘I don’t necessarily understand, but go for it’, or she hasn’t necessarily got the time because of her other, the other demands of her job. But **I know I have her support, and I know that she trusts me and she will back me up.**”*
*Oakwood School, PSE teacher*

In contrast to this, the wellbeing lead in Greenfield School (engagement rank 4, embeddedness rank 3) was not only outside of the SLT, but also had little insight into the process by which wellbeing issues were taken forward within the SLT. They had no direct health and wellbeing-related communication with any SLT members apart from two assistant head teachers. They relied on these most junior members of the SLT to take wellbeing issues forward to SLT meetings, thus perhaps exerting more limited influence on the orientation of the rules and ethos of the school system towards health and wellbeing [8,11]. The wellbeing lead for Greenfield School described a desire for direct communication with the head teacher, demonstrating that she did not perceive this support to be in place. Hence, it may not be the allocation of the role of wellbeing lead to the SLT which matters, so much as support for the role from the SLT.


*“Obviously **the head teacher is a very busy man**. I do speak to him if there’s something like when we had our Healthy Schools Assessment, obviously I spoke to him and if there’s something I usually I wouldn’t necessarily (…). **It would be good to have a specific, like allocated time for that maybe but, I don’t think that’s going to happen**, but it would be good (…)”*
*Greenfield School, wellbeing lead*

Moreover, it was evident that school staff in all case studies perceived a need for more than one individual to be working on health and wellbeing. The comprehensive distributed leadership structure reported by Highbridge School (engagement rank 1, embeddedness rank 1) may have been developed in a response to a high level of need and deprivation (>40% FSM entitlement). 


*“we’ve got some **heavy demands on pastoral care** within the school **so we’ve got a dedicated team for each year group** which includes a pastoral support assistant who’ll look out for the health and wellbeing of each child in their year group.”*
*Highbridge School, wellbeing lead*

Distributed leadership was shown on a smaller scale in Oakwood School (engagement rank 2, embeddedness rank 4). While most schools have a Local Authority employed school nurse who is not based at the school, Oakwood School had a full-time school nurse. The wellbeing lead reported frequent interaction with the school nurse and PSE teacher and rated these as important. This is suggestive of a small team structure, but indicates that whole system orientation towards health and wellbeing has not yet been achieved [8,11]. 

Woodlands School (engagement rank 3, embeddedness rank 2), whose net-maps demonstrated an even more fragmented system suggested that much of the health improvement agenda was driven solely by, and was highly dependent upon, the wellbeing lead assistant head teacher. The wellbeing lead from Woodlands School felt that given the number of roles she was undertaking, and the lack of team structure, it was difficult to prioritise health and wellbeing. This was perceived to be a limiting factor for school health improvement by both the wellbeing lead herself and other members of staff within the school, who expressed a desire for a team structure to be developed


*“certainly a network within school primarily because I feel overwhelmed. **I do feel overwhelmed. I’ve got great colleagues but everybody’s so busy and everybody’s got their own job descriptions, their own priorities** and even within my own role it falls into a pocket sometimes and it’s not, **it hasn’t got the priority on a day to day basis** (…)”*
*Woodlands school, wellbeing lead*

In contrast, the science teacher/head of student voice from Woodlands School (engagement rank 3, embeddedness rank 2) felt that having one wellbeing lead within the SLT with sole responsibility for school wellbeing was sufficient and that ‘too many chefs spoil the broth’.

## 4. Discussion

Social network and qualitative results showed that schools with a higher level of engagement and embeddedness were more likely to have more senior members of staff in key brokerage roles, a higher level of perceived importance of interactions with other key agents with regards to health and wellbeing, a higher number of staff with roles relating to health and wellbeing, and a higher level of embeddedness of outside agencies into school systems.

Consistent with previous education-based studies of teacher collaboration [21], there was substantial heterogeneity between case studies in terms of network brokerage and cliques (i.e., a subset of three or more alters who are all connected to one another, where no other alter is connected to all members) [37]. Social networks are characterised by cliques of similar individuals [40]. The formation of cliques may be problematic when these represent clusters of insular, homogenous groups with limited communication between them. For example, if a group of Physical Education teachers formed a clique related to health and wellbeing, this may impede the embeddedness of health and wellbeing within the school if little brokerage occurred between this clique and the remainder of the school system. However, cliques can serve a functional purpose where sufficient brokerage exists between them [12], such as when they are connected through weak ties [41]. For example, within previous research, one study of three mental health networks showed that integration within small cliques with overlapping links between them was related to network effectiveness [42]. From a complex adaptive systems perspective, the presence of brokerage may facilitate the flow of information and resources throughout a school system, within and between activity settings, supra-systems (such as Education Authorities) and sub-systems (such as classrooms or year groups) [8,11]. In all case study schools, at least some alters in influential brokerage positions (i.e., those with the highest brokerage) were members of the SLT.

Highbridge School reported the highest level of engagement with the research network and embeddedness of health and wellbeing and the most organised distributed leadership structure for health and wellbeing with the head teacher in a key brokerage role. Thus, this may help facilitate engagement with the introduction of diverse information from outside (e.g., research network feedback reports) and the embeddedness of health within a school system [8,36]. Harris [43] identified barriers to distributed leadership in school improvement. These included the need for teachers in formal leadership to relinquish control over the activity and the need to remunerate staff when they take on extra responsibility. 

Distributed leadership has also been shown to increase teacher commitment [44] and sustainability through minimising the negative impact of a specific individual leaving the school [45]. A distributed team structure with non-teaching staff who have dedicated wellbeing roles may help reorient the complex school system towards health and wellbeing by changing the ethos of the school and making health and wellbeing more visible. Furthermore, it may help to improve the efficiency of health and wellbeing-related information flow and facilitate positive change and adaptation in response to a disruption to system functioning caused by the intervention [8,46].

Perceived importance, but not frequency, of health and wellbeing-related interactions were greatest where wellbeing leads were more senior. This could suggest that senior members of staff are more strategic in who they interact with. This may equate to greater access to key decision makers and effect on the efficiency of system functioning and orientation towards health and wellbeing, in line with complex adaptive systems thinking [8,11]. 

Results suggest that senior leadership involvement in collaborating with a wide range of stakeholders may facilitate engagement with a research network and embeddedness of health in the school. Moreover, harnessing these services and creating dedicated non-teaching roles for wellbeing may be key facilitators for creating an embedded team structure for health. This is supported by previous research which found that the provision of evidence summaries and extra support from stakeholders may help to increase action in the form of evidence-based practice [47]. Further support comes from Inchley et al. [48], who conducted a mixed methods process evaluation of two schools attempting to implement an HPS approach. They found that the allocation of responsibility to a member of the SLT helped to embed health and facilitate the delegation of responsibility and liaison with outside agencies. In contrast they found a reliance upon leaders’ commitment and ability to convey enthusiasm to others where the role was allocated to a member of teaching staff outside the SLT [48].

Moore et al. [49] conducted quantitative analyses of school commitment to health, finding no correlation between the allocation of leadership to teaching staff versus a member of the SLT and the implementation of health improvement. However, organisational commitment to health, in terms of SLT overview of health improvement, was substantially correlated with health improvement actions. This implies the importance of support for the wellbeing lead from the SLT, as opposed to the placement of the role within this group [49]. Further research is required to compare and contrast team structures and their impact on system functioning across a larger number of schools.

### Strengths and Limitations

This study employed a cross-sectional design, meaning cause and effect could not be inferred. A key strength of this study is its conceptualisation of schools as complex adaptive systems within an applied research study, employing mixed methods. Qualitative data highlighted the potential for other important determinants of the embeddedness of health and wellbeing within schools, such as the geographical layout of the school, which could be explored further within future research.

Case studies were sampled pragmatically, due to difficulty with recruitment and sampling of staff was undertaken through a pragmatic process with reliance on the wellbeing lead. The participating wellbeing leads were all female, which may have impacted the results. Future research should aim to assess whether and to what extent gender differences occur within the health and wellbeing-related networks of wellbeing leads. Despite this, case study schools did represent a continuum of engagement and other characteristics, offering an in-depth overview of the functioning of varied complex school systems and incorporated a broad range of both positive and negative opinions.

Ego-network analysis, while conferring key advantages over whole network analysis in that it enables interactions beyond the school gates to be captured, also makes assumptions regarding the importance of the wellbeing lead in ensuring the delivery of health improvement within school systems. Embedding ego-network analysis within semi-structured interviews encouraged discussion of how relationships may affect the implementation of health improvement within schools [30,50]. Further qualitative analysis is required to investigate interactions with other systems and sub-systems, such as parents and students, which were reported within the ego network analysis [8,10,11].

## 5. Conclusions

Overall, this study has advanced our understanding of how social networks of school staff and complex school system dynamics may impede or facilitate engagement with intervention, or embedment of health and wellbeing within the school. Allocation of responsibility for wellbeing to a member of the SLT alongside a distributed leadership approach may represent important steps towards the reorientation of school systems towards health and wellbeing. Conceptualising schools as CASs, which respond in diverse ways to the same external stimuli, draws focus to the likelihood that attempting to provide interventions without first engaging with school systems to understand their existing dynamics and how these impede or facilitate health improvement, is likely to give rise to highly variable emergent outcomes. The use of ego-network analysis to understand variance in complex school system starting points, in terms of network structure, could be replicated on a larger scale. This could be utilised to design complex interventions, which work with the system to achieve change.

## Figures and Tables

**Figure 1 ijerph-16-01694-f001:**
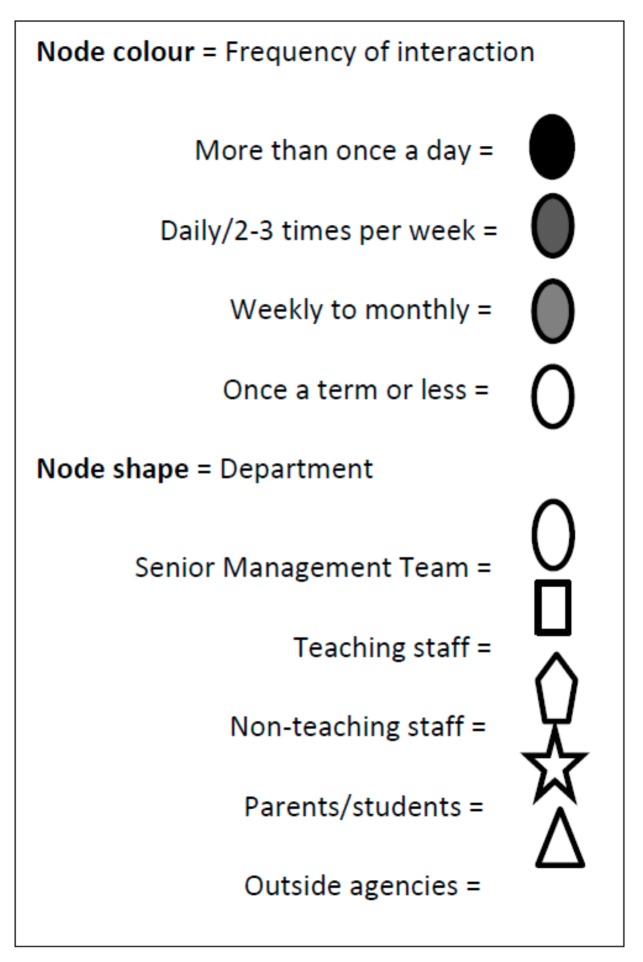
Key for net-map diagrams.

**Figure 2 ijerph-16-01694-f002:**
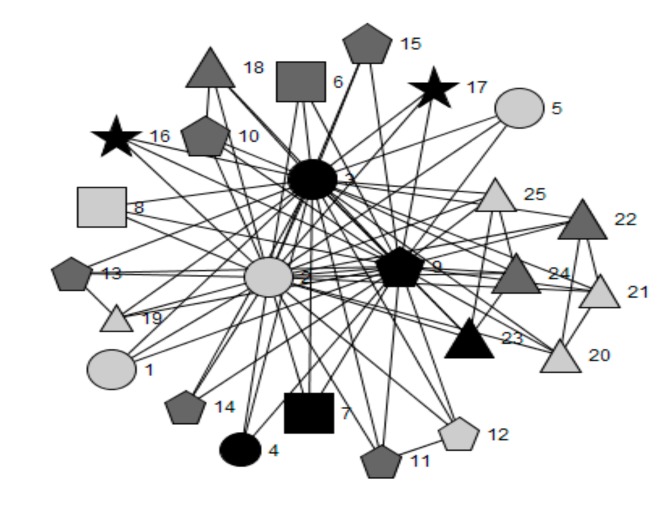
Net-map of wellbeing lead’s ego network for Highbridge School.

**Figure 3 ijerph-16-01694-f003:**
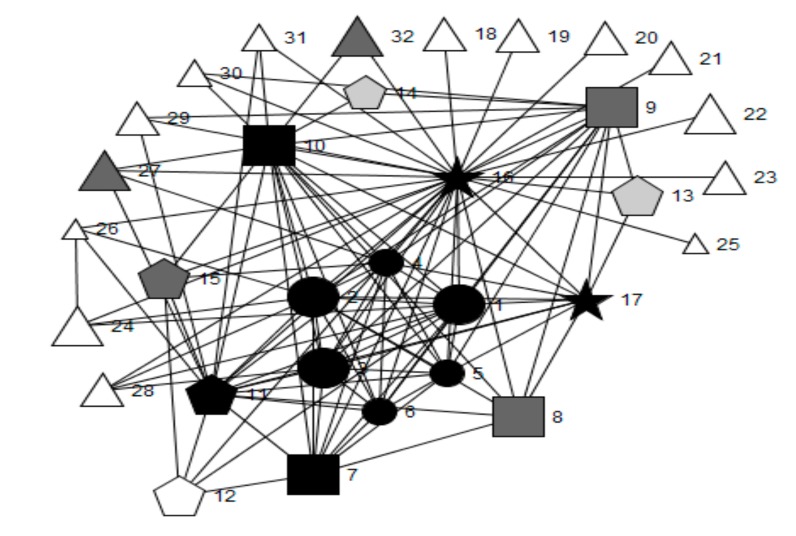
Net-map of wellbeing lead’s ego network for Oakwood School.

**Figure 4 ijerph-16-01694-f004:**
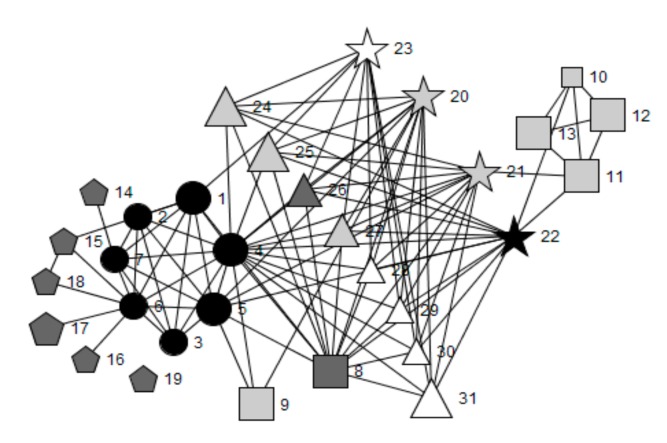
Net-map of wellbeing lead’s ego network for Woodlands School.

**Figure 5 ijerph-16-01694-f005:**
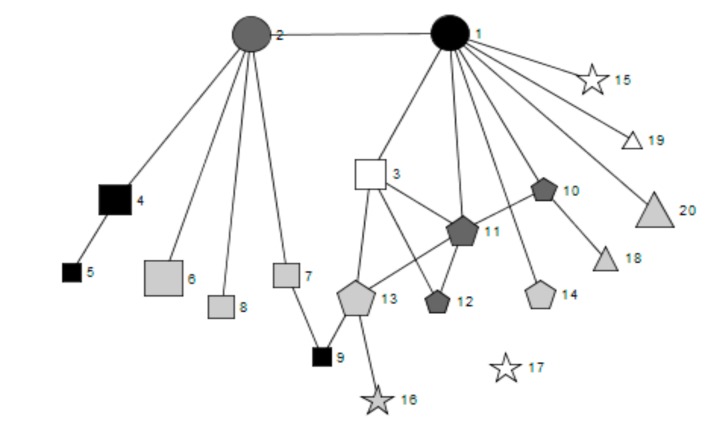
Net-map of wellbeing lead’s ego network for Greenfield School.

**Table 1 ijerph-16-01694-t001:** Study data sources.

**Data Source**	**Date**	**Participants**	**Uses**
Data usage survey	September 2014–June 2015	Wellbeing Leads within 4 case study schools.	To derive contextual measures of school engagement with pupil-level feedback from the School Health Research Network.
Ego Network Analysis	October 2014–April 2015	Wellbeing leads within 4 case study schools.	To measure the characteristics of wellbeing leads’ health and wellbeing networks, and the position of key change agents within this.
Semi-structured qualitative interviews	October 2014–April 2015	4 case study schools (wellbeing leads, members of staff, members of staff and a healthy schools Coordinator at differing positions in the wellbeing leads’ ego-networks.	To explore stakeholder perceptions of wellbeing leads’ health and wellbeing networks, and the position of key change agents within this.
School Environment Questionnaire	March–May 2016	A representative (Wellbeing Lead or a member of senior leadership) from each case study school.	To derive contextual measures of embeddedness of health improvement within case study schools aligned with three topics within the Health Promoting Schools Scheme [2]: Curriculum, environment (measured by a number of policies related to health) and parental involvement.

**Table 2 ijerph-16-01694-t002:** Case study characteristics.

School	No. of Students	Welsh Index of Multiple Deprivation Score (Low Score = Highest Deprivation) *	Geographic Location **	Stage of Health Promoting Schools Scheme ***	Characteristics of Wellbeing Lead	Engagement with the School Health Research Network (Ranking 1–4) ****	Embeddedness of Health Improvement in the School (Out of 3) *****
Greenfield	<900	Highest 10% (affluent)	Rural	National Quality Award (highest accolade)	Female PE Teacher, aged 26–35 years	4	1.66 (rank 3)
Woodlands	>1200	Around median	Welsh Valleys	Stage 1	Female Assistant Head Teacher, aged 46–55 years	3	1.83 (rank 2)
Highbridge	<700	Lowest 10% (deprived)	Urban	National Quality Award	Female Deputy Head, aged 46–55 years	1	2.43 (rank 1)
Oakwood	>1000	Highest 10% (affluent)	Urban	Stage 3	Female Deputy Head, aged 46–55 years	2	1.34 (rank 4)

* The Welsh Index of Multiple Deprivation is a score calculated for each small area of Wales, based on data related to income, employment, health, education, access to services, community safety, physical environment and housing [32]. ** The Welsh Valleys are a unique geographic location and are areas characterised by ex-coal-mining towns and villages and high levels of deprivation. *** These stages range from Stages 1–6 with schools able to be assessed for the highest accolade, the National Quality Award, once they have been a member for 8–9 years [33]. **** Case study schools were ranked highest to lowest according to their level of engagement with the research network and represented a continuum of this engagement from 1 to 4. Further information provided in the text. ***** The composite indicator of embeddedness of health improvement related to the three topics within the HPS Scheme: Curriculum, environment (measured by a number of policies related to health) and parental involvement, resulting in scores of 1 (lowest) to 3 (highest). Further information provided in the text.

**Table 3 ijerph-16-01694-t003:** Characteristics of school staff interviewees.

		Greenfield School	Woodlands School	Highbridge School	Oakwood School
Wellbeing Lead	Role	PE Teacher	Assistant Head Teacher	Deputy Head Teacher	Deputy Head Teacher
	Age group	26–35	46–55	46–55	46–55
	Gender	Female	Female	Female	Female
Interviewee 2	Role	Assistant Head for PSE	Food Technology Teacher	Wellbeing Manager	School Nurse
	Age group	36–45	26–35	36–45	46–55
	Gender	Male	Female	Female	Female
Interviewee 3	Role	Healthy Schools Coordinator	PE Teacher	Behaviour Support Officer	Head of PSE
	Age group	26–35	26–35	36–45	36–45
	Gender	Female	Female	Female	Female
Interviewee 4	Role	Food Technology Teacher	Head of Science and Student Voice	Teaching Assistant	Senior Learning Support Officer
	Age group	36–45	26–35	36–45	46–55
	Gender	Female	Female	Female	Female
Interviewee 5	Role	Student Support Manager			
	Age group	46–55			
	Gender	Female			

**Table 4 ijerph-16-01694-t004:** Top five scores for betweenness centrality for health and wellbeing-related ego networks within each case study (excluding students).

Betweenness Centrality		Highest Scores
Greenfield School	1	Assistant Head (Wellbeing and Safeguarding) (98)
	2	Assistant Head (PSE Line Manager) (71)
	3	Student Support Team (LSAs) (27)
	4	Learning and Wellbeing Department Manager (26)
	=5	Head of PE, Parent-student Support and Head of Student Support (17)
Woodlands School	1	Assistant Head 3 (126)
	2	Deputy Head 1 (87)
	3	All year groups (74)
	4	Assistant Head 4 (45)
	5	Girls’ PE Teacher (36)
Highbridge School	=1	Head Teacher (74)
	=1	Safeguarding Officer (74)
	=1	Wellbeing Manager (74)
	=2	All other alters (0)
Oakwood School	1	Heads of Year (23)
	2	School Nurse (20)
	3	Additional Learning Needs Coordinator (15)
	4	Deputy Head (10)
	5	Head Teacher (7)

**Table 5 ijerph-16-01694-t005:** Characteristics of Wellbeing Leads’ health and wellbeing-related ego networks.

Alter Attribute		Greenfield School	Woodlands School	Highbridge School	Oakwood School
Frequency of interaction between alters and ego	More than once a day	4/20 (20.0%)	8/31 (25.8%)	7/25 (28.0%)	11/32 (34.4%)
	Daily to 2–3 times a week	4/20 (20.0%)	8/31 (25.8%)	9/25 (36.0%)	5/32 (15.6%)
	Weekly-monthly	8/20 (40.0%)	10/31 (32.3%)	9/25 (36.0%)	2/32 (6.3%)
	Once a term or less	3/20 (15.0%)	5/31 (16.1%)	0/25 (0.0%)	3/32 (9.4%)
	Unknown	1/20 (5.0%)	0/31 (0.0%)	0/25 (0.0%)	1/32 (3.1%)
Importance	Not important	3/20 (15.0%)	1/31 (3.2%)	1/25 (4.0%)	2/32 (6.3%)
	Important	5/20 (25.0%)	12/31 (38.7%)	1/25 (4.0%)	5/32 (15.6%)
	Very important	7/20 (35.0%)	11/31 (35.5%)	8/25 (32.0%)	8/32 (25.0%)
	Extremely important	5/20 (25.0%)	7/31 (22.6%)	15/25 (60.0%)	17/32 (53.1%)

**Table 6 ijerph-16-01694-t006:** Number (and percentage) of health and wellbeing-related interactions within each department that have been rated with a high frequency and extreme importance.

Attribute		Senior Leadership Team	Teaching Staff	Non-Teaching Staff	Parents and Students	Outside Agencies
Frequency of interaction >2–3 times per week	Greenfield School	2/2 (100.0%)	3/7 (42.9%)	3/5 (60.0%)	0/3 (0.0%)	0/3 (0.0%)
	Woodlands School	7/7 (100.0%)	1/5 (20.0%)	7/7 (100.0%)	1/4 (25.0%)	1/8 (12.5%)
	Highbridge School	2/5 (40.0%)	2/3 (66.6%)	6/7 (85.7%)	2/2 (100.0%)	4/8 (50.0%)
	Oakwood School	6/6 (100.0%)	4/4 (100.0%)	2/5 (40.0%)	2/2 (100.0%)	2/15 (13.3%)
Interactions rated as extremely important	Greenfield School	2/2 (100.0%)	1/7 (14.3%)	1/5 (20.0%)	0/3 (0.0%)	1/3 (33.3%)
	Woodlands School	0/7 (0.0%)	0/10 (0.0%)	0/6 (0.0%)	4/4 (100.0%)	3/8 (37.5%)
	Highbridge School	4/5 (80.0%)	3/3 (100.0%)	2/7 (28.6%)	2/2 (100.0%)	4/8 (50.0%)
	Oakwood School	3/6 (50.0%)	4/4 (100.0%)	4/5 (80.0%)	2/2 (100.0%)	4/15 (26.7%)

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
