# Peer review of "From Complex Interventions to Complex Systems: Using Social Network Analysis to Understand School Engagement with Health and Wellbeing"

_ijerph, 2019, doi:10.3390/ijerph16101694_

Round 1

Reviewer 1 Report

This paper aims to increase understanding of how network structures of school staff may facilitate or impede attempts to orient school systems toward health and wellbeing. It concludes that allocation of responsibility for wellbeing to a member of the senior leadership team alongside a distributed leadership approach may represent important steps towards the reorientation of school systems towards health and wellbeing.

Used methods, a combination of qualitative and ego social network analyses, are reall pertinent for complex interventions understanding.

I have few minor comments and suggestions that could help authors to improve the quality of the manuscrit.

Methods section:

There is no clear      description of the data collected in qualitive interviews. This information      is partly provided in results section “Data from qualitative interviews      (...) detail below » (P11). This paragraph      should be in methods section.

P6:  “After this,      egos were asked to draw lines between alters to represent whether these alters interacted with      each other in relation to health”. Could you explicit what      “interact in relation to health” refers to (such as giving some examples)?

Results section:

Authors distinguished      different categories of stakeholders. They mentioned one of      them as “senior management team”. Then, they mentioned the “senior      leadership team”. Is it the same category/person?

Please explicit      abbreviations used in the main text.

Discussion section:

Profiles of interactions  between the 4 cases are interesting. The conclusion about the links between the senior leadership involvement, the collaboration with a wide  range of stakeholders and the engagement with embeddedness of health in      the school could be more qualified.  Authors made methodological precautions in “limits” section. But they      could bring also other differential explanations other than  the      effect of the position of the wellbeing leads in schools. For example, the      interactions between stakeholders, and hence the engagement with      embeddedness of health, may be influenced by some other factors such as:      individual characteristics (charisma, seniority, etc), occupational      environment/organization (e.g. matches between stakeholders’ timetables, the area layouts, etc).      These factors could be independent from the position per se or  moderators of it. I suggest to discuss it. 

In this study, all the      wellbeing leads are female. Maybe male and female could be different in      the way to interact, the importance of this interaction, the choice of      other alters, etc, limiting the transferability of these conclusions for      male wellbeing leads? This could be discussed. 

In a knowledge transfer      perspective, it could be interesting to illustrate the interactions within      a clique with a      concrete example. It could clarify the direct interest of this analysis  for agents in charge of implementing health strategies in school.

Author Response

The authors would like to thank the reviewers for their constructive comments. Since submitting, we have added a small number of recent references to the background literature. We present a response and an outline of the edits made to the manuscript below. 

Reviewer 1:

This paper aims to increase understanding of how network structures of school staff may facilitate or impede attempts to orient school systems toward health and wellbeing. It concludes that allocation of responsibility for wellbeing to a member of the senior leadership team alongside a distributed leadership approach may represent important steps towards the reorientation of school systems towards health and wellbeing.

Used methods, a combination of qualitative and ego social network analyses, are reall pertinent for complex interventions understanding.

I have few minor comments and suggestions that could help authors to improve the quality of the manuscrit. 

Methods section:

There is no clear      description of the data collected in qualitive interviews. This information      is partly provided in results section “Data from qualitative interviews      (...) detail below » (P11). This paragraph      should be in methods section.

This paragraph has been moved to the methods section.

P6:  “After this,      egos were asked to draw lines between alters to represent whether these alters interacted with      each other in relation to health”. Could you explicit what      “interact in relation to health” refers to (such as giving some examples)?

The following examples have been added; Examples of such interactions include planning a new health improvement activity, such as a new physical activity programme or sharing information about pupils experiencing issues, such as poor mental health. 

Results section:

Authors distinguished      different categories of stakeholders. They mentioned one of      them as “senior management team”. Then, they mentioned the “senior      leadership team”. Is it the same category/person?

Please explicit      abbreviations used in the main text.

This has been changed to senior leadership team throughout, apart from where the term senior management team is mentioned within participant quotes.

Discussion section:

Profiles of interactions  between the 4 cases are interesting. The conclusion about the links between the senior leadership involvement, the collaboration with a wide  range of stakeholders and the engagement with embeddedness of health in      the school could be more qualified.  Authors made methodological precautions in “limits” section. But they      could bring also other differential explanations other than  the      effect of the position of the wellbeing leads in schools. For example, the      interactions between stakeholders, and hence the engagement with      embeddedness of health, may be influenced by some other factors such as:      individual characteristics (charisma, seniority, etc), occupational      environment/organization (e.g. \matches between stakeholders’ timetables, the area layouts, etc).      These factors could be independent from the position per se or  moderators of it. I suggest to discuss it. 

These issues did arise within the qualitative data and, therefore, highlight a strength of this mixed methods study. This has been highlighted in the strengths and limitations section, with a call for future research to explore these issues further.

In this study, all the      wellbeing leads are female. Maybe male and female could be different in      the way to interact, the importance of this interaction, the choice of      other alters, etc, limiting the transferability of these conclusions for      male wellbeing leads? This could be discussed. 

This has been added to the strengths and limitations section, with a call for future research to explore these issues further.

In a knowledge transfer      perspective, it could be interesting to illustrate the interactions within      a clique with a      concrete example. It could clarify the direct interest of this analysis  for agents in charge of implementing health strategies in school.

The definition of a clique has been repeated within the discussion and the following example has been added. ‘For example, if a group of Physical Education teachers formed a clique related to health and wellbeing, this may impede the embeddedness of health and wellbeing within the school if little brokerage occurred between this clique and the remainder of the school system.’

Reviewer 2 Report

The paper has an important justification of the study, highlights the convenience of using social networks methods and understanding how health improvement information flows through school systems. The limitations of the study are well indicated and should be taken into account by the readers.

The authors express the importance of the issue:“ Social dynamics within and between a systems’  activity settings may have the potential to improve or harm health” and the influence of the school environmental on the studies “Within schools, interpersonal interactions  and relationships are shaped not only by individual characteristics of agents (e.g. preferences, choices, and motivations), but also by characteristics and institutional practices of the school”.

The objective of the paper is “to measure wellbeing  leads’ ego networks and determine the extent to which health and wellbeing roles are embedded into school systems. Next, semi-structured qualitative interviews are used to explore the broader school  context surrounding these network structures. This paper will aim to increase understanding of how variability in network structures, and the positions of key change agents within these, may facilitate or impede attempts to orient school systems toward health and wellbeing” (p.139).

However, the manuscript has some weakness, which must be improve:

p.44Indeed, the “minimally disruptive” nature of health education, in terms of impact on school  systems, is perhaps why they have been favoured to date, though perhaps simultaneously why they have failed to disrupt entrenched patterns of health and inequality" [7].I recommend not to write exactly the same, literal. We can found this exactly literal reference in this thesis (See p. 250): https://orca.cf.ac.uk/100889/2/Hannah%20Littlecott_Final%20version%20thesis%20with%20corrections%2011.5.17.pdf

Review the reference : (7) Hawe, P. Lessons from complex interventions to improve health. Pub Health 2015;36:307. 

Author Response

The authors would like to thank the reviewers for their constructive comments. Since submitting, we have added a small number of recent references to the background literature. We present a response and an outline of the edits made to the manuscript below. 

The paper has an important justification of the study, highlights the convenience of using social networks methods and understanding how health improvement information flows through school systems. The limitations of the study are well indicated and should be taken into account by the readers.

The authors express the importance of the issue:“ Social dynamics within and between a systems’  activity settings may have the potential to improve or harm health” and the influence of the school environmental on the studies “Within schools, interpersonal interactions  and relationships are shaped not only by individual characteristics of agents (e.g. preferences, choices, and motivations), but also by characteristics and institutional practices of the school”.

The objective of the paper is “to measure wellbeing  leads’ ego networks and determine the extent to which health and wellbeing roles are embedded into school systems. Next, semi-structured qualitative interviews are used to explore the broader school  context surrounding these network structures. This paper will aim to increase understanding of how variability in network structures, and the positions of key change agents within these, may facilitate or impede attempts to orient school systems toward health and wellbeing” (p.139).

However, the manuscript has some weakness, which must be improve:

p.44Indeed, the “minimally disruptive” nature of health education, in terms of impact on school  systems, is perhaps why they have been favoured to date, though perhaps simultaneously why they have failed to disrupt entrenched patterns of health and inequality" [7].I recommend not to write exactly the same, literal. We can found this exactly literal reference in this thesis (See p. 250): https://orca.cf.ac.uk/100889/2/Hannah%20Littlecott_Final%20version%20thesis%20with%20corrections%2011.5.17.pdf

Review the reference : (7) Hawe, P. Lessons from complex interventions to improve health. Pub Health 2015;36:307. 

This sentence has been reworded as follows, ‘While adding health topics to the curriculum is relatively straightforward, more complex multi-level interventions have often proven challenging to implement [7], perhaps tending to achieve minimal disruption to school system functioning, and entrenched patterns of health and inequality, as a consequence [8].’

Reviewer 3 Report

This paper presents a method for understanding variations in school engagement with health and being. The authors conducted ego network analysis with 4 staff, representing 4 case study schools, identifying their school health improvement. Data was supported by qualitative interview data from staff identified through these networks. Although this is a small study, the methodology has broader application. The authors conclusion is valid, that “Conceptualising schools as complex adaptive systems, which respond in diverse ways to the same external stimuli, draws focus to the likelihood that attempting to provide interventions without first engaging with school systems to understand their existing dynamics and how these impede or facilitate health improvement, is likely to give rise to highly variable emergent outcomes. The use of ego-network analysis to understand variance in complex school system starting points, in terms of network structure, could be replicated on a larger scale.”

Overall, the manuscript is very well written and makes a contribution to the literature on network analysis applications in public health. The background pertaining to this study is comprehensive, the results are self-explanatory, and the discussion draws out relevant points. However, I have several (generally minor and addressable) questions regarding Methods.

Line 110. Further clarify the term health improvement, i.e. whether school policy, curriculum, Health & Safety, mental, physical, etc. This is a critical term used throughout the manuscript.

Table 1. This table should be more comprehensive. We read later that background school data was collected in 2014-2015, in addition to the school environment questionnaire in 2016, for ranking parameters of the 4 case study schools.

Line 164. Define the school health research network

Line 164. Are there relevant similarities/differences, for this study purposes, between the schools within the health research network?

Line 164. Four from 77 invited schools participated. How many schools were planned to recruit?

Table 2. This table needs to be referenced after information on engagement and embeddedness is provided.

Table 2. Explain stages of health promoting schools

Table 2. Include footnotes for each criterion

Line 171. Replace the word thesis with manuscript/research.

Line 175. Is the current study of 4 schools embedded within a broader study that warranted conducting a survey, inviting all schools within the research network, plus collecting additional background data 1-2 years later from 100 schools (line 185)? Explain.

Line 176. Explain what ‘pupil-level feedback from the research network’ means

Line 189-199. Has the indicator for embeddedness been validated?

Line 215. Minor point. The word ‘each’ is missing between within and case.

Line 219. Was the ego network researcher led or conducted by staff independently?

Line 226. Was the ego network verified by the participant after analysis?

Line 239. Table 3 indicates there were 4-5 interviewees, whereas line 239 states 3-5. These should agree.

Table 3. Define PSE

Line 255. Include the initial coding system that was developed

Line 277 onwards. Emphasise that the networks are of health improvement. E.g. ‘a highly organised team structure’ of who interacts with who in relation to health improvement, from the ego’s perspective.

Figures 2-5. Which node is the ego that derived the schematics showing their perspective the of the health improvement network? It would have been interesting to compare ego networks from the perspective of the alters.

Tables 4-6. These should all state the nature of the metrics (i.e. health improvement)

Author Response

The authors would like to thank the reviewers for their constructive comments. Since submitting, we have added a small number of recent references to the background literature. We present a response and an outline of the edits made to the manuscript below. 

This paper presents a method for understanding variations in school engagement with health and being. The authors conducted ego network analysis with 4 staff, representing 4 case study schools, identifying their school health improvement. Data was supported by qualitative interview data from staff identified through these networks. Although this is a small study, the methodology has broader application. The authors conclusion is valid, that “Conceptualising schools as complex adaptive systems, which respond in diverse ways to the same external stimuli, draws focus to the likelihood that attempting to provide interventions without first engaging with school systems to understand their existing dynamics and how these impede or facilitate health improvement, is likely to give rise to highly variable emergent outcomes. The use of ego-network analysis to understand variance in complex school system starting points, in terms of network structure, could be replicated on a larger scale.”

Overall, the manuscript is very well written and makes a contribution to the literature on network analysis applications in public health. The background pertaining to this study is comprehensive, the results are self-explanatory, and the discussion draws out relevant points. However, I have several (generally minor and addressable) questions regarding Methods.

Thank you. 

Thank you for this positive review. We have outlined our edits below.

Line 110. Further clarify the term health improvement, i.e. whether school policy, curriculum, Health & Safety, mental, physical, etc. This is a critical term used throughout the manuscript.

The following definition of the term health improvement has been added near the beginning of the introduction (lines 9-10): ‘School health improvement activity can be defined as actions, such as policies and practices, employed with the aim of improving the health and wellbeing of students (Littlecott et al., 2018).’

Table 1. This table should be more comprehensive. We read later that background school data was collected in 2014-2015, in addition to the school environment questionnaire in 2016, for ranking parameters of the 4 case study schools.

More detailed information has been added, including a column for date of data collection and a row for the data usage survey.

Line 164. Define the school health research network

A definition has been added where it is first mentioned (line 160-164).

Line 164. Are there relevant similarities/differences, for this study purposes, between the schools within the health research network?

These data were used only to provide context, thus analyses did not go into this detail.

Line 164. Four from 77 invited schools participated. How many schools were planned to recruit?

We planned to recruit 4 case study schools and sampled from the 69 members of the School Health Research network employing Purposive sampling using replication logic to ensure that case studies were varied in terms of geographical location, size and socioeconomic status.

Table 2. This table needs to be referenced after information on engagement and embeddedness is provided.

These references have been added.

Table 2. Explain stages of health promoting schools. Table 2. Include footnotes for each criterion

A description of the Welsh Index of Multiple Deprivation, geographic location, stages of the Health Promoting Schools scheme in Wales, engagement with the School Health Research Network and embeddedness of health improvement in the school have been added as footnotes under Table 2. 

Line 171. Replace the word thesis with manuscript/research.

This has been replaced.

Line 175. Is the current study of 4 schools embedded within a broader study that warranted conducting a survey, inviting all schools within the research network, plus collecting additional background data 1-2 years later from 100 schools (line 185)? Explain.

Information has been added to the sections ‘case study engagement with the School Health Research Network’ and ‘Embeddedness of health improvement within case study schools’ to clarify that the data used to contextualise case study schools within this study were collected by the School Health Research Network and not as part of the current study.

Line 176. Explain what ‘pupil-level feedback from the research network’ means

The School Health Research Network conducts biannual surveys with member secondary schools in Wales and provides individualised wellbeing reports to schools outlining statistics relating to their pupils’ levels of diet, physical activity, substance use, mental health and wellbeing. This has been made clear within the following sections; ‘case study sampling’ and ‘case study engagement with the School Health Research Network’.

Line 189-199. Has the indicator for embeddedness been validated?

This is not a validated measure. The text does already state that this was ‘created by the research network team’. Line 196-197. To our knowledge, no validated measure exists for embeddedness of health. We have used this measure in the following previous paper; Littlecott, H.et al. 2018. Health improvement and educational attainment in secondary schools: complementary or competing priorities? Exploratory analyses from the School Health Research Network in Wales.Health Education and Behavior45(4), pp. 635-644.

Line 215. Minor point. The word ‘each’ is missing between within and case.

The word ‘each’ has been added.

Line 219. Was the ego network researcher led or conducted by staff independently?

This was researcher led and embedded within the qualitative interviews. This has been clarified at the beginning of the section, ‘Qualitative interviews with wellbeing leads and other school staff’. Line 252-253.

Line 226. Was the ego network verified by the participant after analysis?

No. The network analysis took place within an interview whereby interviewees created a visualisation of their network and talked around this. We used Egonet software to create a computer-generated version of this exact network.

Line 239. Table 3 indicates there were 4-5 interviewees, whereas line 239 states 3-5. These should agree.

This has been altered to four to five.

Table 3. Define PSE

Personal and Social Education (PSE) has been defined within the following section, ‘Qualitative interviews with wellbeing leads and other school staff’. Line 257-258.

Line 255. Include the initial coding system that was developed

The initial overarching themes were as follows; comparison with other schools, data usage, family’s role in health promotion, friends’ role in health promotion, health and wellbeing programmes in the school, interactions, job role, link between health and educational outcomes, perceptions of school health promotion, school ethos and social network analysis. These have been added to the section ‘Qualitative interview analysis’. Lines 279-283.

Line 277 onwards. Emphasise that the networks are of health improvement. E.g. ‘a highly organised team structure’ of who interacts with who in relation to health improvement, from the ego’s perspective.

Above each net-map, it has been emphasised that these health and wellbeing-related networks were reported from the perspective of the wellbeing lead in each school. 

Figures 2-5. Which node is the ego that derived the schematics showing their perspective the of the health improvement network? It would have been interesting to compare ego networks from the perspective of the alters.

The net-map diagrams visualise the individuals that the wellbeing lead or ‘ego’ has reported interacting with. Thus, the wellbeing lead is not included within the net-map as they are linked to every node. This has been outlined within the section, ‘Influential champions for health: Characteristics and position within social networks’. Lines 303-305.

Tables 4-6. These should all state the nature of the metrics (i.e. health improvement)

‘Health and wellbeing-related’ has been added to the title of tables 4-6.